# Experimental Study on Physical Behavior of Fluidic Oscillator in a Confined Cavity with Sudden Expansion

**Hadi Samsam-Khayani, Shabnam Mohammadshahi**  **and Kyung Chun Kim** *

School of Mechanical Engineering, Pusan National University, Busan 46241, Korea;
hadisamsam@pusan.ac.kr (H.S.-K.); shabnam@pusan.ac.kr (S.M.)
* Correspondence: kckim@pusan.ac.kr; Tel.: +82-51-510-2324

**Abstract:** In this study, two-dimensional time-resolved particle image velocimetry (2D-TR-PIV) was used to investigate the effect of the external domain on oscillating jets from double-feedback fluidic oscillators. Two different cases with different Re numbers (2680–10,730), as free external domain and fully confined were studied. Time-averaged results showed although a self-oscillating jet was attained for the free external domain, it could not be achieved for a fully confined geometry. For a fully confined geometry at Re = 2680, two symmetric vortices did not allow the jet to oscillate and at Re = 6440, the flow pattern in the external region became non-symmetric due to the Coanda vortex, subsequently, the self-oscillating jet was not observed. At Re = 10,730, the strength of the jet was inclined to cope with such vortices and tended to oscillate. However, strong vortices were created near the exit region of the fluidic oscillator, which led to an almost non-symmetric pattern. In addition, the proper orthogonal decomposition (POD) method and phase-averaged analysis were applied to obtain the unsteady behavior of flow and the most energetic dynamic structure. Interestingly, at Re = 6440, the third mode was still energetic for fully confined, but for other cases, the first two modes were the most energetic modes, which showed vigorous coherent structures.

**Keywords:** fluidic oscillator; experimental measurement (TR-PIV); proper orthogonal decomposition (POD); phase averaged; fully confined

## 1. Introduction

In the literature, the jet behavior for a variety of free and bounded geometries were investigated considerably. A multitude of studies is devoted to investigating the emitting jet in a 2D channel. This kind of jet in a cavity might be considered as a flow that can produce and generate self-sustaining oscillations. Based on the geometry, self-sustaining oscillations could coarsely be classified as jet edge, hole tone, organ type, sudden expansion, jet confined in a cavity, and fluidic oscillator (Figure 1).

The confinement's effect on the entrainment of the confined self-sustaining oscillating jet is broadly appealing and has enticed scholars to investigate that. Self-sustaining phenomena are carried out in a wide range of Reynolds number from 100 [1–4] up to 170,000 [5], which leads to the conclusion that in both laminar and turbulent jets, this mechanism can be attained.

In addition, for different ranges of Reynolds numbers, several studies about a jet in the sudden expansion were reported. Cherdron et al. [6] by using the laser-Doppler anemometry, experimentally analyzed the velocity characteristics of asymmetric flow for low Reynolds number (110–1000), which forms in symmetric sudden-expansion geometry. Fearn et al. [7] numerically and experimentally studied the flow at Reynolds numbers in the range of 25–380 and described that the flow becomes time-dependent at higher Reynolds numbers. Casarsa and Giannattasio [8] investigated the turbulent

flow through the sudden expansion at Re = 10,000 by performing 2D-PIV measurements. They observed an asymmetric flow and concluded that the flow structure is slightly sensitive to the variation of the Reynolds number in the considered range.

So far, there are no studies to explain the unique characteristic of an oscillating jet from a double feedback fluidic oscillator, which is emitted in a 2D channel with sudden expansion. Thus, in the current work, the oscillating jet emitting in a fully confined external domain (Figure 2a), is compared with the free oscillating jet. Fluidic oscillators are widely used as actuators for active flow control, because these devices exert a high momentum and a wide range of frequency, they do not have any moving part, and have a simple design. Based on the number of feedback loops, fluidic oscillators are divided into three main groups—feedback-free, single-feedback, and double-feedback jets. The most popular type is double-feedback channels, as illustrated in Figure 2b.

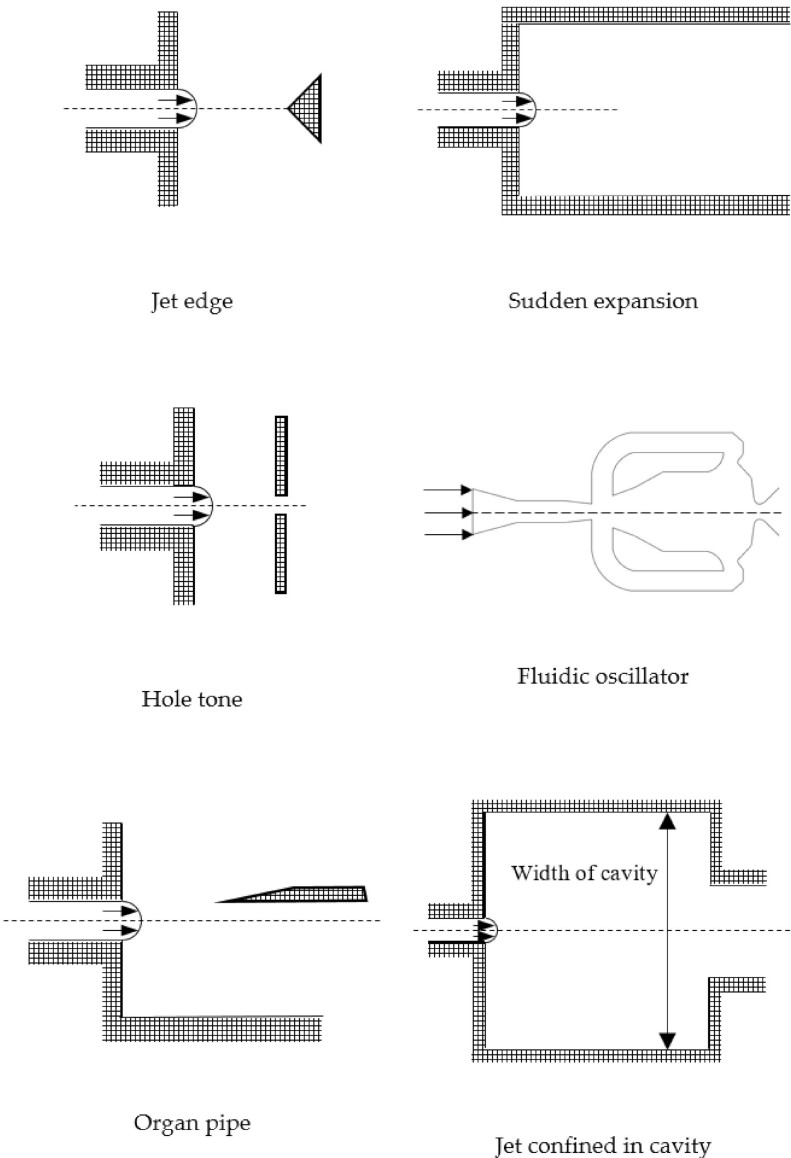

Jet edge

Sudden expansion

Hole tone

Fluidic oscillator

Organ pipe

Jet confined in cavity

**Figure 1.** Classification of self-sustained oscillator.

When the supply fluid enters the mixing chamber, it attaches to one side of the chamber because of the Coanda effect [9]. One part of the fluid goes out to the external domain, while the other part near the exit is diverted through the feedback channel and impinges on the main jet at the inlet. This part causes the jet to attach to another side of the mixing chamber, and the process is done continuously

with a specific frequency. Therefore, an oscillating jet is emitted to the external domain, without using mechanical parts. In recent decades, this type of fluidic oscillator attracted much research interest and is used in various applications, like heat transfer enhancement [10–16], mixing enhancement [17–21], and separation control [22–24].

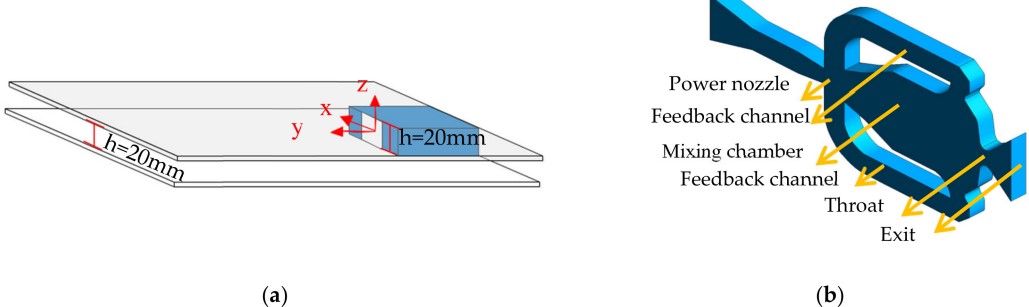

(**a**)                                   (**b**)

**Figure 2.** (**a**) Fully confined external domain, and (**b**) fluidic oscillator with double-feedback channel.

In the literature, it is alleged that the oscillation frequency of the jet is dependent on the internal region. For the first time, Bobusch et al. [25] experimentally investigated the internal part of a fluidic oscillator. Then, some numerical and experimental studies were also conducted to identify the mechanisms that drive the jet to oscillate with a certain frequency [25–29]. They concluded that by increasing the mass supply for the fixed geometry, the frequency increases. In numerous studies, researchers focused on the external domain of the fluidic oscillator and discussed the impact of the oscillating jet, relative to the spreading angle, which is controlled by the jet flow rate [30–35].

Critical appraisal of the lack of investigation of the external domain's effects on the oscillating jet from double feedback fluidic oscillator showed that further study is required. In this study, we portrayed the confined external domain as an influential parameter to be considered for future design and development of the application of oscillating jets. Thus, we concentrate on an oscillating jet emitted in a fully confined, with sudden-expansion geometry and compare with a free oscillating jet, at Reynolds numbers from 2680 to 10,730, while the fluidic oscillator geometry is fixed. An oscillating jet in sudden-expansion flow is complicated by the nature of the interaction between the shear layers and the Coanda effect, which is caused by the confinement of the external region's channel. One compelling point that persuades us to conduct this work is that experimental results contradict other studies discussed above and show the jet does not oscillate like the free external domain oscillating jet, due to confinement effect, as well as while the external domain has steps in the normal direction to the emitted flow. Additionally, the oscillating jet can become asymmetric, despite symmetric test sections, and this phenomenon should be considered, especially in some applications like heat and mass transfer enhancement or flow control. For instance, efficient cooling of gas turbines is required [36,37], and if the effective area of the cooling is not enough, the blades can melt, and the heat from surface blades cannot be removed. Since 2D-TR-PIV is paving the way as a utilitarian tool to investigate flow structure, it is used in this study. Furthermore, Proper Orthogonal Decomposition (POD) and phase-averaged analysis were applied to the velocity field to find the most energetic dynamical structures and unsteady behavior.

## 2. Experimental Setup and PIV Measurement

We looked at a double-feedback fluidic oscillator that had a rectangular throat cross-section ($10 \times 20$ mm$^2$), hydraulic diameter ($D_h$) of 13.3 mm, and a ratio of the width of the jet throat to the depth equal to $AR_j = 0.5$. The exit cross-section was $40 \times 20$ mm$^2$, and the diverging angle of the nozzle was $90°$. The confinement of the external region was investigated, which was obtained by inserting two plates at the bottom and the top of the external domain of the fluidic oscillator, resulting in a channel

with the same height as the fluidic oscillator's exit. Then, it did not take a step in the z-direction, as shown in Figure 3a.

2D-2C TR-PIV was widely used to measure the flow velocity field in a plane and was applied to study the flow structures of the fluidic oscillator. A water tank that was made with acrylic, which had the same refractive index as water, was seeded with silver-coated hollow glass spheres with an average diameter of 11.7 μm and a density of 1.1 g/cm$^3$. Since there was no significant discrepancy between the densities of these particles and the water, these particles could readily follow the water without any destruction of the flow.

The experimental setup and PIV system consists of a continuous laser (532-nm wavelength and 5 W), a cylindrical lens with a focal length of 1000 mm, and a spherical lens with a focal length of 25 mm were used to attain a laser sheet less than 1 mm thickness to minimize the three-dimensional effects on the PIV measurement, and a high-speed camera with a resolution of 1024 by 1024 pixels (Photron FastCam SA1.1), and a lens with an f-number of 2.8 (Nikon, micro lens 65 mm), as shown in Figures 3 and 4. In addition, post-processing of PIV images was conducted by software developed by our group, to extract the velocity fields from sequence images. A cross-correlation scheme was performed to obtain the instantaneous and ensemble-averaged velocity vector and a final interrogation window of $16 \times 16$ pixels with 50% overlap was applied to satisfy the Nyquist criterion. The uncertainty in the velocity vector, based on the size of the interrogation window, sampling rate, and particle density (6–8 seeding particles in each interrogation window), was estimated to be 2% [38]. The flow rate of the jet was controlled by a flowmeter. Based on the hydraulic diameter and mean velocity ($U_j = Q/A_{throat}$), the Reynolds number spanned the range of 2680–10,730.

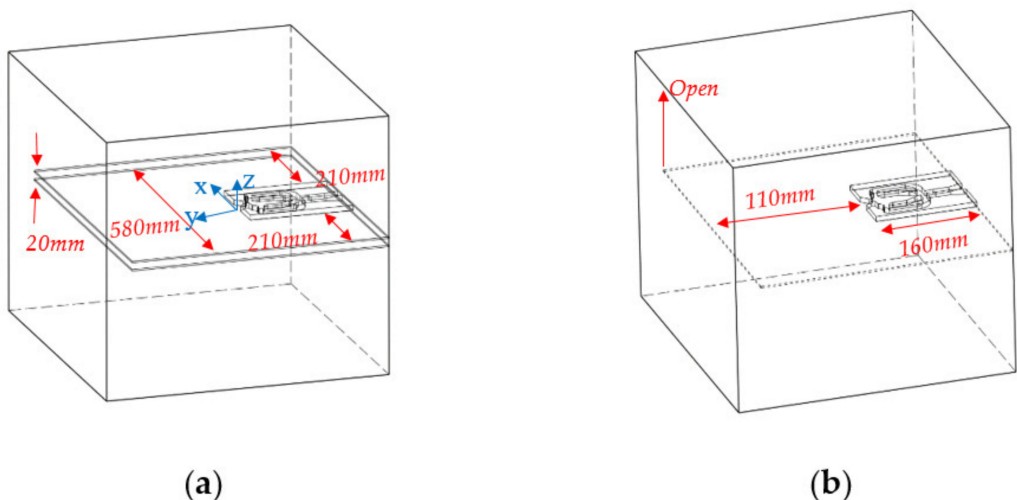

**Figure 3.** Diagram and size of (**a**) fully confined and (**b**) free geometry.

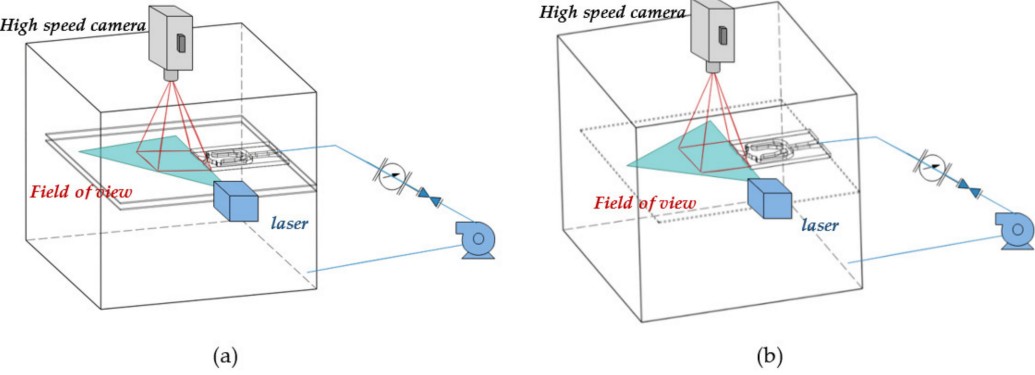

**Figure 4.** Experimental setup for (**a**) fully confined and (**b**) free external domain.

## 3. Results and Discussion

### 3.1. Time-Averaged Results

The velocity signals of two symmetrical points at the centerline of the jet were extracted to attain the frequency of the sweeping jet in the free external domain. Two points with a 180° phase lag were selected, so the quality of the signals could be enhanced. By using a low-pass filter and Fast Fourier Transport (FFT) analysis on the subtracted velocity signal, the oscillation frequency was achieved, as shown in Figure 5. The frequency of the sweeping jet in the free external domain at each Reynolds number was dependent upon the mass flow rates and the internal geometry. As illustrated in Figure 5, the frequency of the oscillating jet for the free external domain, approximately increased linearly with Reynolds number, because of the increment of the mass flow rate.

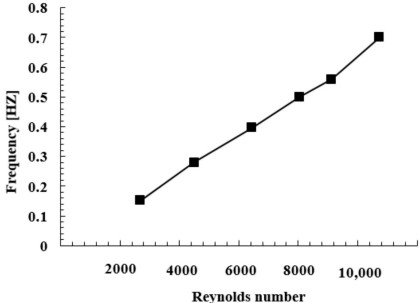

**Figure 5.** Frequency versus Reynolds number for free external domain (the uncertainty was estimated at around 2% [38]).

### 3.1.1. Velocity Distribution

The non-dimensional time-averaged velocity field for the sweeping jet in the free external domain is shown in Figure 6. As the results show, by increasing the Reynolds number, the spreading angle of the oscillating jet was enlarged for Re= 2680 to Re = 6440 and reached a maximum deflection of 90 degrees at Re = 6440. This meant that the flow was fully attached to the nozzle exit's diverging wall. Thus, by increasing the Reynolds number from 6440 to 10,730, the maximum deflection angle was almost constant because, for this range, the deflection angle was identical to the nozzle exit angle. However, as Re increased, the inhomogeneous velocity distribution became clearer, and the weak flow within the maximum concentration of the flow disappeared. Additionally, as Kim et al. [30]. suggested, by increasing Re, the jet spent more time at its maximum deflection because of the Coanda effect, and the oscillating jet did not move with a constant oscillating speed. Due to the Coanda effect, a recirculation bubble formed within the mixing chamber, which was a charge of the oscillating pattern at both the internal and external regions. Moreover, by increasing the Reynolds number, the size of the recirculation bubbles grew. Therefore, the spreading angle and frequency of the oscillating jet increased and a different flow pattern was created in the external region.

The velocity ratio was defined as the maximum value of the non-dimensional v-component to the maximum value of the non-dimensional u-component. The results of the sweeping jet in the free external domain in Figure 6 show that by increasing the Reynolds number, this ratio increased from 0.46 to 1 for Re = 2680 to Re = 10,730. Therefore, for low Reynolds numbers, there is a strong streamwise velocity and weak transverse velocity of the continuous jet, with a small spreading angle jet. For the highest Reynolds number, the velocity ratio was increased, and the spreading angle grew to the maximum value (90°). Thus, as the jet flow rate increased, the jet momentum changed from an even distribution at the centerline for Re = 2680 to being mainly distributed on the lateral sides, which caused a V-shaped time-averaged velocity field. Thus, for the fixed geometry in the current work, the Strouhal number, $St = f\, D_h/U_j$, was constant, since the oscillation frequency changed linearly with the velocity.

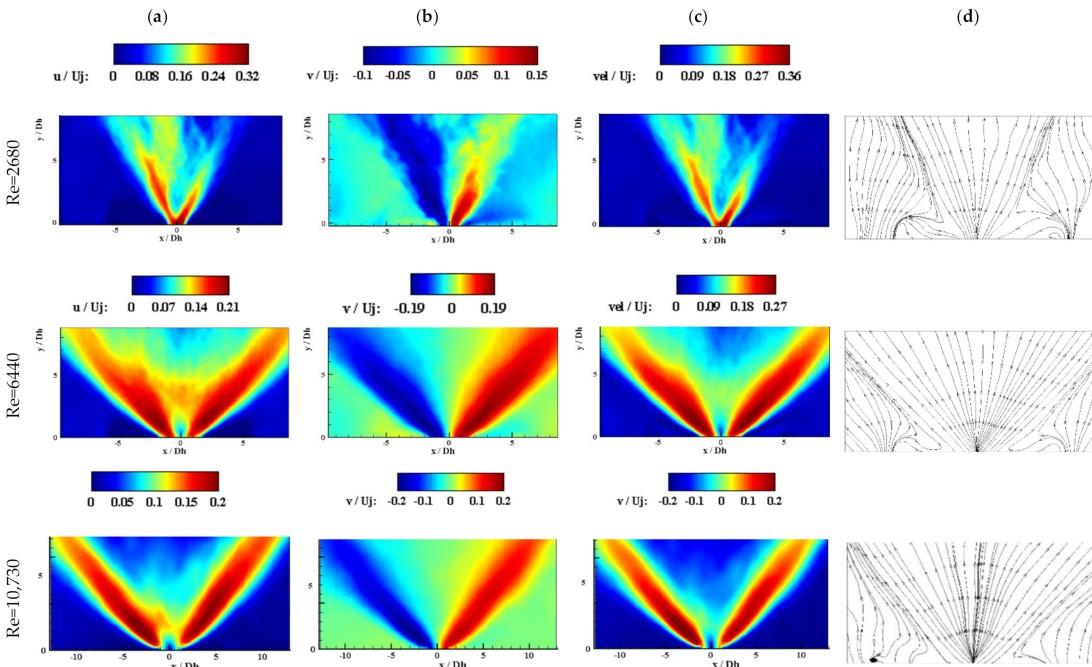

**Figure 6.** Time average of (**a**) streamwise velocity, (**b**) transverse velocity, (**c**) velocity magnitude, and (**d**) streamline for free external domain at Re= 2680, 6440, and 10,730 and $z/D_h = 0$.

To investigate the effect of confinement on the flow pattern and the performance of sweeping jet, the non-dimensional time-averaged velocity field for the fully confined external domain is depicted in Figure 7. Although a sweeping jet pattern was produced and controlled by the internal part (double-feedback fluidic oscillator), the jet did not oscillate at this range of Reynolds number as it did for the sweeping jet in the free external domain. This phenomenon was opposite to the results of the previous study, while the external domain had steps in the normal direction to the emitted flow [31], where notable differences between the effect of confinement and Re number could be concluded.

First, in the fully confined domain, at the lowest Reynolds number 2680 where the jet could not oscillate similar to the sweeping jet in the free external domain, the jet emitted to the cavity and created two identical recirculation zones, and due to the shear stress at the boundary of the jet, two vortices with equal sizes were created. Thus, by incrementing the Reynolds number to 6440, the shear stress at the boundary of the jet increased. Due to the Coanda vortex and different pressure distribution across the jet, the jet was inclined to attach to one side. However, the internal side was oscillating due to the effect of the Coanda vortex and the feedback loop (or recirculation zone) within the mixing chamber, but the jet could not oscillate at the cavity because the momentum or the strength of the jet was not enough to cope with the power of the vortex, shear stress, and vortex's suction toward the boundary of the jet. Indeed, Section 3.2. describes this phenomenon in more detail, which states that the frequency of the oscillating jet at the external domain was around 0.001 [Hz], while for the internal domain it was calculated to be around 0.4 [Hz]. Furthermore, when the recirculation zone at one side of the cavity was created, due to the small diameter and high velocity, it led to a high-pressure shortage in this recirculation area. This caused the jet to be pulled toward the one side wall, which led to supply the large-pressure shortage and more deflection of the jet. Moreover, when the jet reached the sidewall, further shrinking of the recirculation area could not be achieved. Thus, when the increment of the shortage of pressure in this recirculation area diminished then the pressure increased. Subsequently, the other recirculation area at the cavity started to be stronger than before, which caused the jet after a while to attach to the opposite side of the wall, which showed the non-ergodicity pattern in the cavity. The compelling point was that the frequency of this oscillation at the cavity that illustrated the shifting from the left to the right side walls of the external domain, in Re = 6440, equaled to 0.001046 [Hz],

which was smaller than the internal part and the sweeping jet in the free external domain, both of which were almost 0.4 [Hz]. Thus, it could be postulated that the jet's oscillation was independent of the time at the external region or cavity.

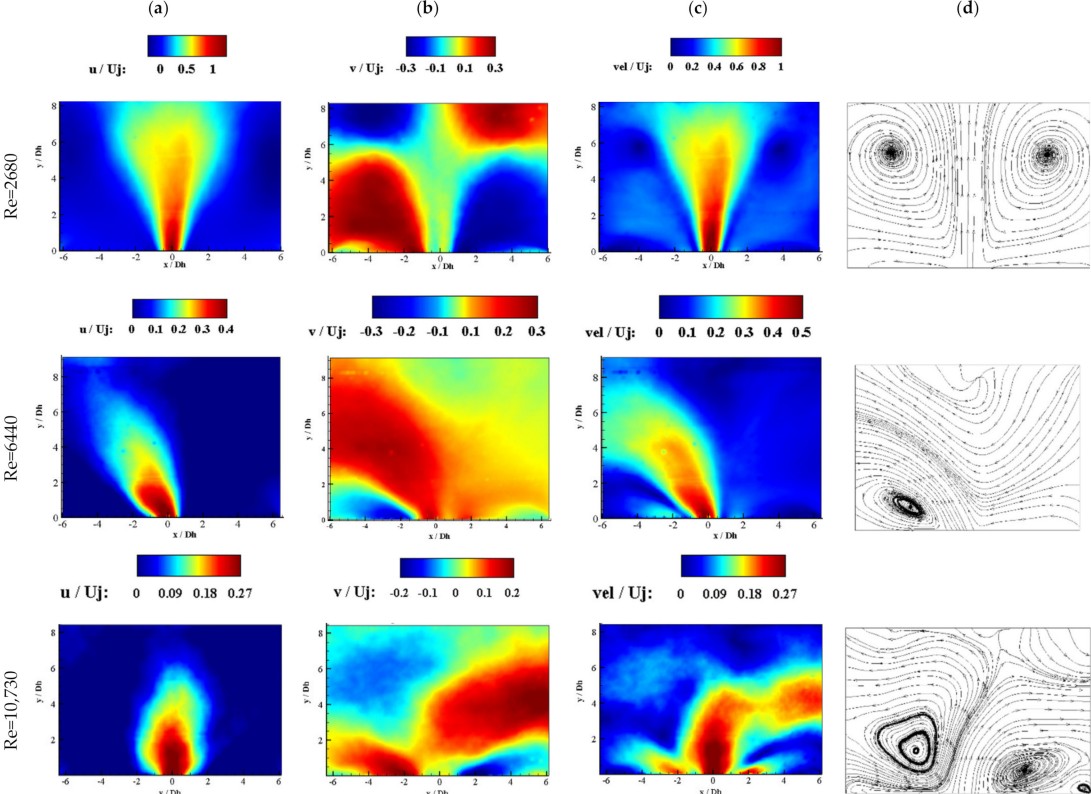

**Figure 7.** Time average of (**a**) streamwise velocity, (**b**) transverse velocity, (**c**) velocity magnitude, and (**d**) streamline for fully confined external region at Re= 2680, 6440, and 10,730 and $z/D_h = 0$.

For Re = 10,730, the jet was generated by the fluidic oscillator and changed the direction from right to left, so the final time-averaged pattern was inclined toward the centerline with the highest velocities. However, the jet was not symmetrically distributed, but almost a steady and continuous jet was obtained in the time-averaged results (Figure 7).

Additionally, the fully confined geometry had a higher maximum value of the non-dimensional velocity distribution at a constant flow rate or Re number for both the u-component and the v-component (Figure 7). This occurred because the volume of the surrounding fluid was higher for the free external domain case, and the jet core was weaker due to the entrainment flow towards the jet core. For the free external domain, the flow was entrained from the direction normal to the oscillation pattern, and this entrainment increased, while the contact area between the jet and ambient fluid increased by increasing the flow rate. In contrast, for the fully confined geometry, the entrainment was prevented, and some strong vortices were created, which affected the flow pattern, as illustrated in Figure 7.

For Re = 2680, two vortices were created on both sides of the jet, which led to a smaller spreading angle in comparison to the free external domain, as shown in Figure 7. By increasing Re to 6440, the jet could not oscillate freely and attached to the left side wall of the cavity because the vortex became strong and pulled the jet toward it (because of the shear layer, some small vortices were created and accumulated together, which led to a large vortex). In fact, there was no entrainment flow toward the external domain from the normal direction. Subsequently, to satisfy the continuity equation, the vortex was inclined to circulate continuously. This led to the control of the flow in the external region and did not allow the flow to oscillate. The strength of the oscillating jet was not enough, and the V-shaped pattern could not be produced. This phenomenon was not favorable for heat transfer enhancement

because the film cooling was not complete, and some surfaces of the blades can melt due to the lack of appropriate film cooling.

For the highest Re number, Re = 10,730, some oscillating patterns could be seen, and vortices that were created at the Re = 6440, by increasing the momentum of the jet, could be detached from the sidewall. As the figures show, the distance of the vortex's center from the sidewall for Re = 10,730 was higher than Re = 6440. However, the spreading angle and the influenced area were still small in comparison with the free external domain case.

Figure 8 shows the instantaneous velocity field for Re = 6440, at which the time-averaged patterns for two geometries were significantly different. For the free external domain, the jet was moving between its maximum deflection (during T/2). In contrast, for the fully confined geometry, the vortices created in the recirculation zone were strong and did not let the jet oscillate periodically. Therefore, the maximum deflection of the jet was lower for the fully confined geometry. To give a more comprehensive understanding of the flow structures of the fully confined geometry, a video file of the velocity magnitude contours was given as the Supplementary Materials. It should be noted that all figures for fully confined case represent the jet, while it was attached to the left side wall, however, for confirming the non-ergodicity behavior of flow at the external domain, the jet was inclined to the right side wall in video files.

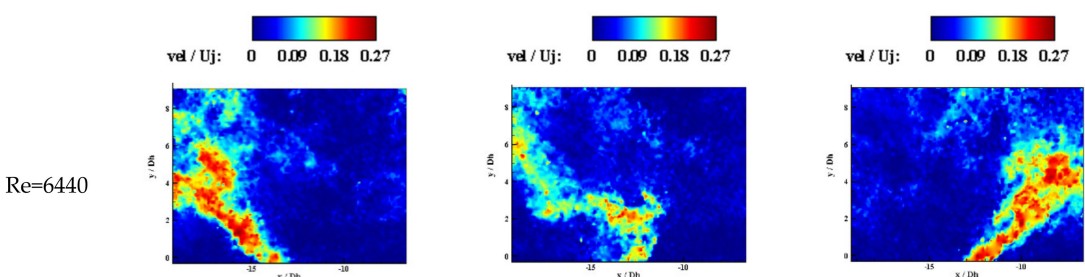

**Figure 8.** Instantaneous non-dimensional velocity magnitude for Re = 6440 in the free external domain.

### 3.1.2. Q-Criterion Distribution

In order to get more details of vortex identification, the Q-criterion was calculated based on time-averaged results (Equation (1)). When Q was positive, vortex regions occurred, and the magnitude of vorticity was greater than that of the rate of strain. Then, the circulation (Equation (2)), was calculated over an area with a positive value of the Q-criterion, as this value was located in the vortex area.

$$Q = \left(|\Omega|^2 - |S|^2\right)/2 \tag{1}$$

$$\Gamma = \iint_A \Omega.dA \tag{2}$$

where $\Omega$ is vorticity, S is strain rate, $\Gamma$ is circulation, and A is the area of the positive Q-criterion.

The mean magnitude of circulations was calculated and listed in Table 1. For the free external domain, the vortices cancelled out each other, and the circulation for the whole domain was almost zero. However, for the fully confined case, the circulation was increased about 10 times, compared to the free external domain. In addition, a linear relation between the Reynolds number and non-dimensional circulation ratio $(DU_j/\Gamma)$ was found. In contrast, this ratio for the free external domain case was almost constant and independent from the Re number.

Time-averaged Q-criterion contours for a better understanding of the large coherent structure distribution are shown in Figure 9. For the free external domain, the maximum concentration of the Q was located at the maximum deflection of the jet. However, for the fully confined external domain, this behavior could be found in the vortex position, as was illustrated in Figure 7d.

**Table 1.** Circulation ($\Gamma$) and $DU_j/\Gamma$ over the area with positive Q-criterion (Q > 0).

| | $\Gamma \times 10^{-3}$ | | $DU_j/\Gamma$ | |
| --- | --- | --- | --- | --- |
| | **Free External Domain** | **Fully Confined** | **Free External Domain** | **Fully Confined** |
| Re = 2680 | 1.815 | 13.19 | 3.467 | 0.214 |
| Re = 6440 | 2.031 | 13.28 | 3.339 | 0.510 |
| Re = 10,730 | 2.751 | 13.90 | 4.109 | 0.813 |

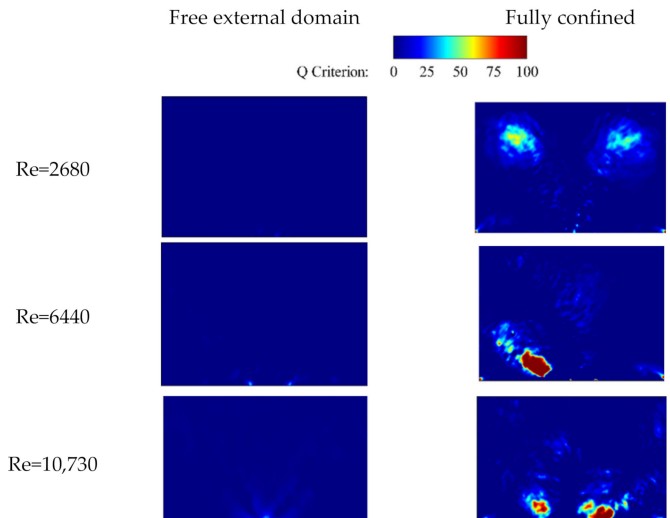

**Figure 9.** Time-averaged Q-criterion contour of free and fully confined external domain at different Reynolds number.

### 3.1.3. Velocity and Turbulence Fluctuation's Profiles

To examine the distribution of the oscillating jet in more detail, time-averaged velocity components and turbulence fluctuations were extracted along a line that was $1D_h$ away from the exit of the fluidic oscillator (Figure 10). The non-dimensional u-component and the velocity magnitude had a small spreading angle in the central region for Re = 2680, and the majority of the jet momentum was in the streamwise direction, along the *y*-axis. As Re increased, these values were reduced quickly to less than half of the peak value. In contrast, for the v-component profiles, there was a reduction of the peak values. Thus, for higher Re numbers, a pair of peaks was created, and the values of the peak almost increased with Re. The turbulence fluctuation's profiles were also similar to the velocity magnitude profiles.

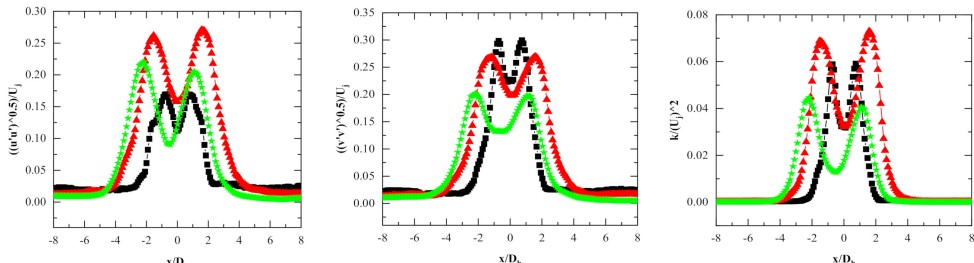

**Figure 10.** Time-averaged velocity components and fluctuations along $y/D_h = 1$ for the free external domain.

The profiles of non-dimensional velocity and turbulence fluctuation for fully confined geometry are shown in Figure 11 along the line $y = 1D_h$. The jet momentum did not have peaks on the lateral sides, unlike the free external domain (Figure 10), and did not switch from the left position to the right position in the external domain in the cavity. For the lowest Reynolds number, Re = 2680, a steady

and continuous jet was generated, and as the Re increased, the behavior of the jet was changed, and it attached to the left wall. For instance, at Re = 6440, the peak value of the non-dimensional velocity magnitude was at $x/D_h = 1$. For Re = 10,730, a jet with a single peak was obvious, which showed a streamwise and continuous jet instead of a V-shaped pattern.

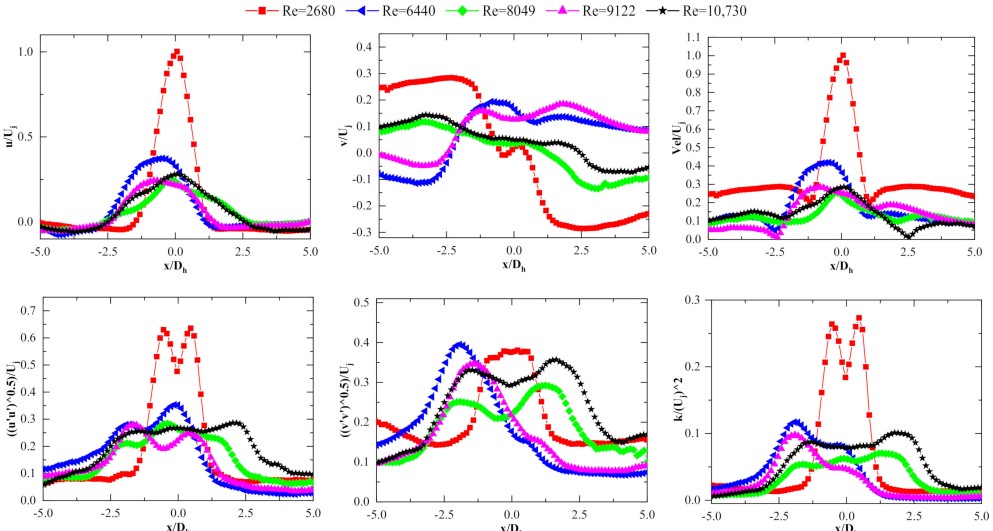

**Figure 11.** Time-averaged velocity components and fluctuations along $y/D_h = 1$ for the fully confined case.

### 3.1.4. Momentum Flux

In order to perform a thorough investigation on the entrainment of fluid, the distribution of momentum flux was conducted. The momentum flux equation could be written as:

$$M_f = \iint V\rho(V.n)dA \rightarrow M_f = \int_{-b/2}^{b/2} \rho u^2 dy \tag{3}$$

where $b$ is the domain's height at the external region. Figure 12 depicts that for the free external domain at Re = 2680, the momentum flux distribution along the external domain was almost constant. Since, the potential of the main jet, compared to the volume of fluid at up, down, and sidewalls were little, it caused smaller entrainment. In contrast, by confining the external domain, the momentum flux increased, compared to the free external domain, because the volume of the surrounded fluid decreased, which led to the potential of the main jet to be able to cope with the small volume of the surrounded fluid. Moreover, when the fluidic oscillator was emitted to the confined domain at Re = 2680, it created two identical vortices at the end of the jet, which caused a reduction in the energy of the core jet by friction and shear layers. At Re = 6440 for the free external domain, the potential of the jet could overcome the volume of the environment fluid, so the momentum flux increased, then decrease at the end of the jet, which had a minimum potential. On the other hand, confinement of the domain led to creating vortices at sidewalls that did not allow the jet to oscillate. Subsequently, the momentum flux decreased due to both the effect of the confined geometry and strong vortices. At Re = 10,730, for the free external domain, the momentum flux increased, and then the entrained fluid reduced the potential of the flow, as well as the momentum flux. On the contrary, the confined geometry led to create strong vortices and a small volume of the surrounded fluid that was conducive to decreasing the momentum flux.

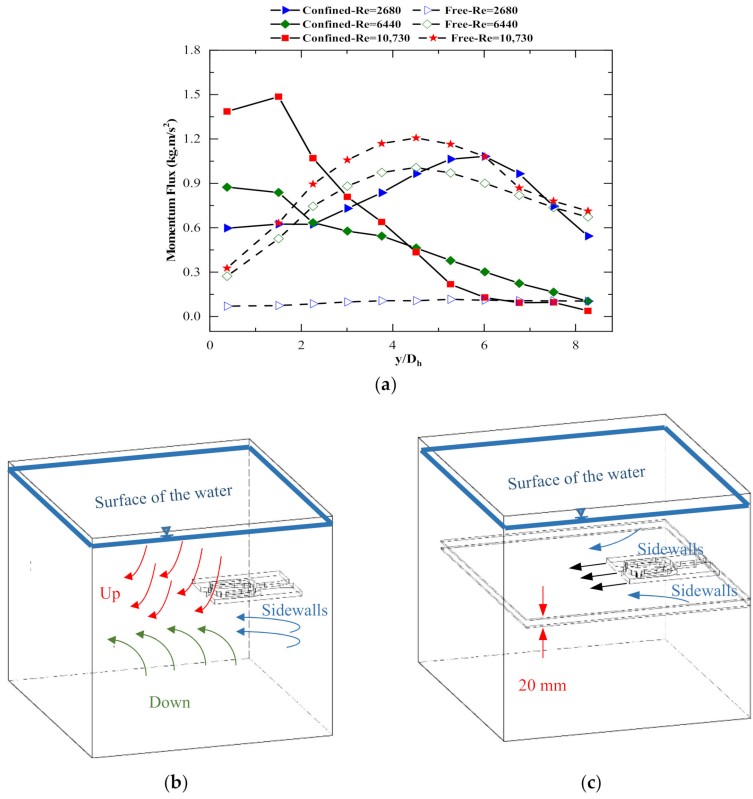

**Figure 12.** (**a**) Momentum flux for both the free and fully confined domains at different Reynolds number, entrainment of fluid for (**b**) free and (**c**) fully confined external domain.

### 3.2. Proper Orthogonal Decomposition (POD) Results

To study the dynamics of the unsteady flow structures, POD was applied to extract major coherent flow structures from an unsteady flow field by extracting the most energetic eigenmodes. The velocity field could be written as:

$$u(x,t) = \overline{u} + u'(x,t) \tag{4}$$

$$u'(x,t) = \sum_{i=1}^{N} a_i(t)\phi_i(x) \tag{5}$$

where $u(x,t)$ is the instantaneous velocity, $\overline{u}$ is the time-averaged velocity, and $u'(x,t)$ is the velocity fluctuation that can be shown by the linear combination of the products of POD modes ($\phi_i(x)$) and corresponding modal coefficients ($a_i(t)$) for each snapshot. The phase angle for each snapshot was calculated by corresponding time coefficients $a_1(t)$ and $a_2(t)$ and for obtaining the phase-averaged fields, the phase angle of $\pm 3°$ was achieved by dividing the entire cycle into 120 phase intervals.

$$\theta = \text{archtan}\left(\frac{a_1(t)}{a_2(t)}\right) \tag{6}$$

The POD analysis included 12,000 velocity field snapshots, and the eigenvalues of POD modes converged to stable constants. The convergence method was also used to find the number of POD modes, as Hekmati et al. [39] proposed. Figure 13 shows the percentage of energy contained in the first 20 modes. The third mode was still energetic at Re = 6440 with around 10% of the energy. In contrast, for the other cases, the first two modes contained a dominant percentage of the total energy and the third mode had less than 5%. On the other hand, for the sweeping jet in the free external domain, as the previous study [25,27] suggested, the first two modes had a predominant percentage of the energy.

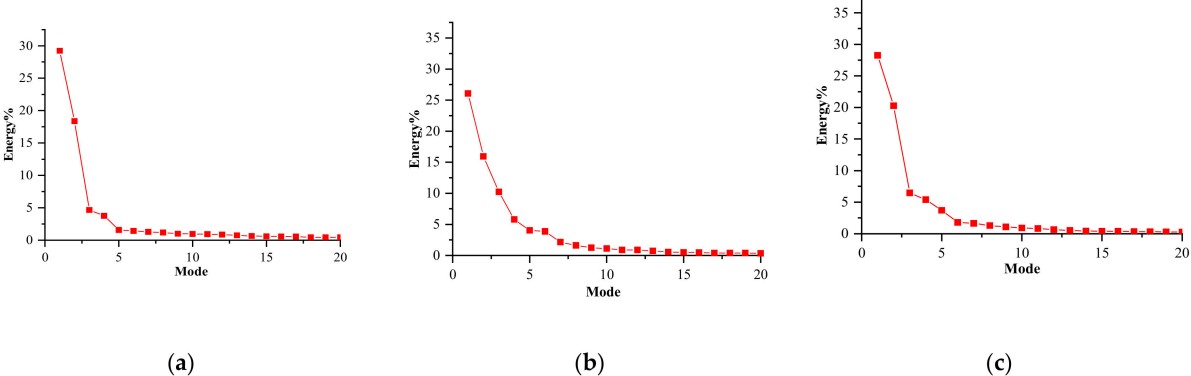

**Figure 13.** POD distribution of energy for 20 modes of the fully confined case at (**a**) Re = 2680, (**b**) Re = 6440, and (**c**) Re = 10,730.

The streamwise and transverse components of the first and second POD modes for both free and fully confined external domains are illustrated in Figures 14–16. A general view of the free external domain shows a similar distribution, with alternating positive and negative velocity fluctuations and symmetrical distribution around the central line. As Re increased, the spreading angle reached its saturation value, which was clear for Re = 6440 and 10,730.

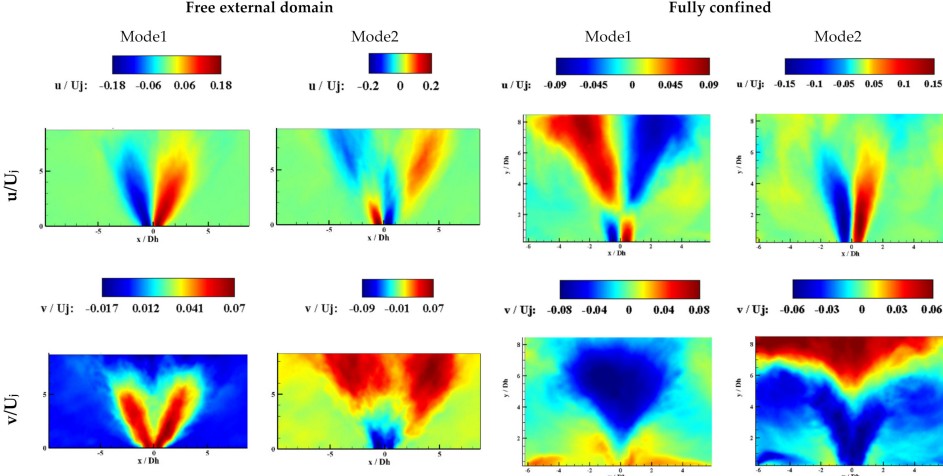

**Figure 14.** Spatial patterns of the POD modes for the both free and fully extrnal domains at Re = 2680.

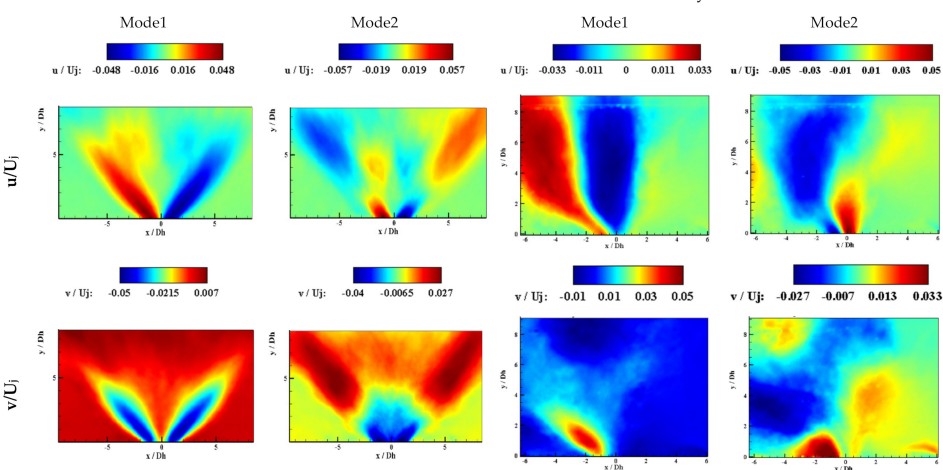

**Figure 15.** Spatial patterns of the POD modes for the both free and fully external domains at Re = 6440.

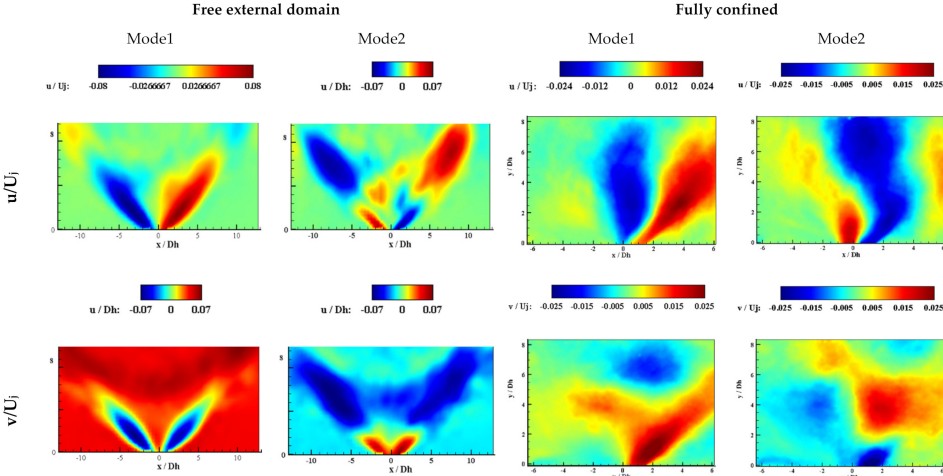

**Figure 16.** Spatial patterns of the POD modes for the both free and fully external domains at Re = 10,730.

For the fully confined geometry, at Re = 2680, the jet was close to the centerline. By increasing the Re number, the values of the non-dimensional velocity components were higher than that of the free external domain. For Re = 6440 and 10,730, as was expected, the high fluctuation areas were on the left and right sides of the cavity, as the jet was attached to those areas and was prevented from periodically oscillating.

Figures 17 and 18 show the time coefficients for the free and fully confined cases at Re = 6440. Figure 17 depicts the circle shape for the amplitude relations of the time coefficients. Thus, the first two modes for both external domains had a phase shift $\pi/2$ and illustrated switching behavior at the outside region. FFT analysis on the temporal time coefficients is listed in Table 2 and showed that the characteristic frequency of time coefficients was identical to the frequency of the sweeping jet for the free external domain (Figure 5), which represents that the first two modes could be aggregated into a pair. Additionally, the distribution of time coefficients revealed that for the fully confined geometry, although the jet could not oscillate as the free external domain case, it oscillated in a small region, while it was inclined toward the sidewall due to the oscillatory behavior at the inside of the fluidic oscillator. Indeed, for the fully confined case, when the jet was attached to one side, there were some dominated patterns that portrayed oscillating behavior and the frequency of those oscillatory patterns was a bit larger than the free external domain, because the oscillating area was reduced, as shown in Figure 18. One compelling point about the POD modes was the frequency of the third mode for the fully confined was unrelated to the first two modes, because as mentioned in Figure 15, it still had a significant percentage of energy. However, the frequency of the fourth POD mode was twice that of the first two modes, which illustrated that the jet passed through this mode twice during each cycle. This relation for the free external domain could be observed for the first three modes that the frequency of the third mode was twice of the first two modes.

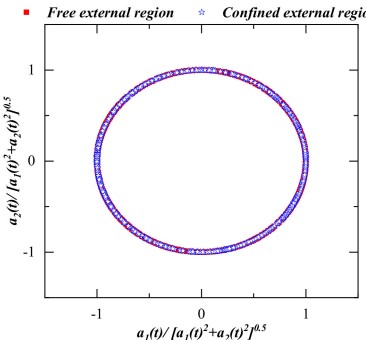

**Figure 17.** Scatter distribution of time coefficients $a_1(t)$ vs. $a_2(t)$ at Re = 6440.

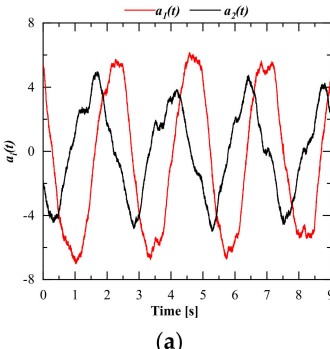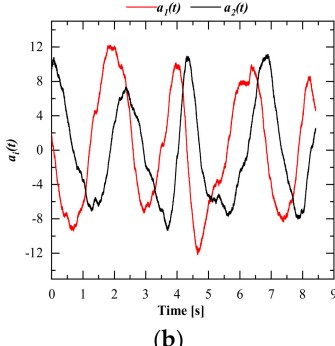

(a)    (b)

**Figure 18.** Corresponding POD time coefficient $a_i(t)$ for both the free and fully confined external domain at Re = 6440; (**a**) free (**b**) fully confined.

**Table 2.** Frequency [Hz] of time coefficient for both the fully confined and free external domain at Re = 6440.

|          | **Free External Domain** | **Fully Confined** |
|----------|--------------------------|--------------------|
| $a_1(t)$ | 0.3968                   | 0.4883             |
| $a_2(t)$ | 0.3968                   | 0.4883             |
| $a_3(t)$ | 0.824                    | 0.2442             |
| $a_4(t)$ | 0.824                    | 0.9156             |

To substantiate the oscillatory pattern and large-scale coherent structures for both free and fully confined external domain, the phase identifications were calculated and depicted in Figures 19 and 20. Phase $\theta = 0°$ for the free external domain described the flow when the jet inside the fluidic oscillator passed through the mixing chamber and attached to one of the converging exit's wall. Thus, the phase $\theta = 180°$ represents the jet while attached to the opposite converging exit's wall and almost mirrored $\theta = 0°$ around the centerline. Phases $\theta = 90°$ and $\theta = 270°$ represent the jet while it reached its maximum deflection. However, for the fully confined domain case, there was no symmetric behavior of the jet around the centerline, or in its oscillating domain at the small area. Indeed, from phase 0° to 180°, the vortex was pushed from the wall near the exit of the fluidic oscillator, subsequently at $\theta = 270°$, it shrunk, due to an increase in the pressure drop and increment in the size of the vortex at the right side of the jet. In addition, the jet almost reached maximum deflection within its oscillating area, at $\theta = 270°$.

Table 2 shows that the frequencies of temporal coefficients for both the free and fully confined external domains are of the same order. Since inside the fluidic oscillator was the charge of the oscillating pattern for the free external domain, we named the cycle of confined geometry, listed in the Table 2, as a "fluidic oscillator cycle" with high frequency. Eventually, when the jet impinged to one of the sidewalls, the left-side vortex could not be further shrunk and the pressure drop began to reduce at the left side, which led to detach the jet and push it to the opposite sidewall and this oscillating pattern was re-initiated. However, the frequency of this cycle was much smaller than the "fluidic oscillator cycle", subsequently, it was named as "cavity cycle" with low frequency. Indeed, based on the experiment, the frequency of the "cavity cycle" was around 0.001046 [Hz].

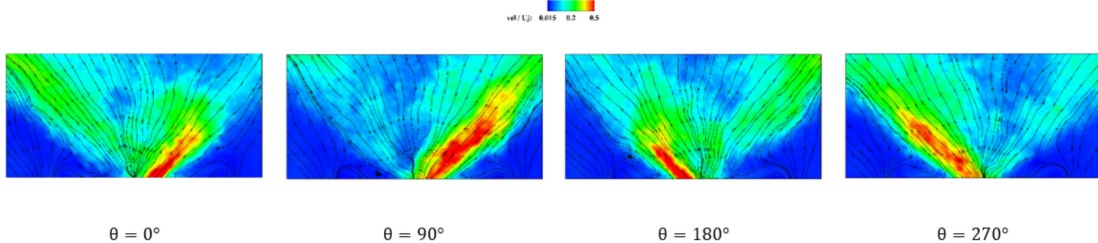

Figure 19. Phase identification for the free external domain at Re = 6440.

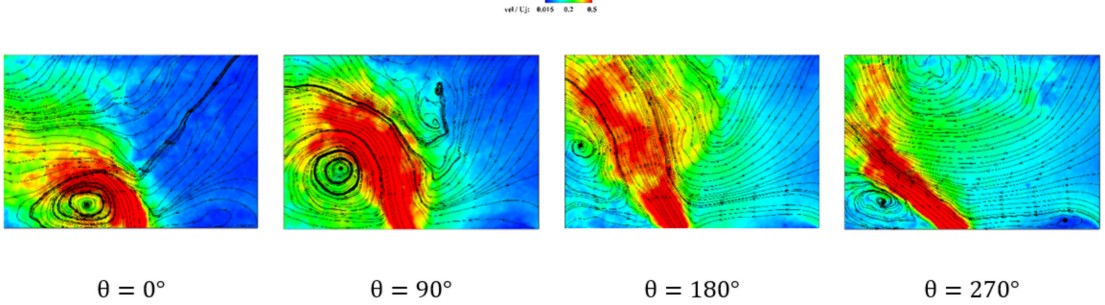

Figure 20. Phase identification for the fully confined external domain at Re = 6440.

## 4. Conclusions

In the current study, the effect of the external domain of an oscillating jet emitted from a double-feedback fluidic oscillator in a quiescent flow was studied via 2D-TR-PIV. Two different geometries were carried out, fully confined, and free external domain. In addition, the Re number was increased to investigate the effect of the flow rate on the flow pattern. For the free external domain, the frequency of the jet was changed linearly by the Reynolds number, and the jet spreading angle was also increased till it reached its saturated value. However, for the fully confined geometry, the velocity distribution pattern was completely different. In order to get unsteady flow characteristics, the POD method and phase-averaged analysis were conducted, and coherent structures were obtained. Some of the significant results were as follows:

- Confinement of the external domain led to create two identical vortices at the lowest Re number and caused the jet to have a smaller spreading angle than the free external domain.
- For Re = 6440, a V-shaped pattern was created in the free external domain, but for the fully confined geometry, the presence of a vortex at the exit region of the fluidic oscillator prevented the formation of a periodic flow, and the jet bent toward it.
- For the fully confined geometry at Re = 10,730, the potential of the main jet tended to cope with the strength of the created vortices, and it was expected that by increasing the Re number, an oscillating pattern would be observed.
- A linear relation between the Reynolds number and the non-dimensional circulation ratio ($DU_j/\Gamma$) exists for the fully confined case. In contrast, this ratio for the free external domain is almost constant and independent from the Re number.
- For the fixed internal geometry, a linear relation between the Reynolds number and oscillation frequency leads to the Strouhal number, St = $f\, D_h/U_j$, being constant.
- POD analysis reveals the fact that for the fully confined case, the third mode was still energetic at Re = 6440, and the first three modes stood for the large coherent flow structures. However, for the free external domain, the first two modes were energetic and showed the periodic behavior of the oscillating flow.
- Phase-averaged analysis showed that for the fully confined case, although the jet was attached to the sidewalls, there was an oscillating pattern at its small oscillation area, with high frequency.

**Supplementary Materials:** The following are available online at http://www.mdpi.com/2076-3417/10/23/8668/s1. Video S1: A movie of instantaneous velocity contour measured by the time-resolved PIV at Re = 6440 with fully confined external domain. This phenomenon is a non-ergodic process due to Coanda effect.

**Author Contributions:** H.S.-K.: Visualization, Conceptualization, Methodology, Software, Investigation, Writing—review & editing. S.M.: Formal analysis, Investigation. K.C.K.: Supervision, Funding acquisition, Writing—review & editing. All authors have read and agreed to the published version of the manuscript.

**Funding:** This work was funded by the [National Research Foundation of Korea (NRF)] grant number [No. 2020R1A5A8018822, No. 2018R1A2B2007117], which is funded by the [Korean government (MSIT)].

**Conflicts of Interest:** The authors declare that they have no known competing financial interests or personal relationships that could influence the work reported in this paper.

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
