# Peer review of "Experimental Study on Physical Behavior of Fluidic Oscillator in a Confined Cavity with Sudden Expansion"

_applsci, doi:10.3390/app10238668_

Round 1

Reviewer 1 Report

This article describes an experimental study comparing the behavior of an oscillating jet with and without external confining walls. The differences among the cases considered are significant and the results are of interest. The quality of the results and the images are good. The write-up needs some polishing.

1. Line 177, You suggest that for the confined case, the internal side is oscillating but the measured external side is not. However, it would seem plausible that the suppression of the oscillation observed externally also corresponds to no internal oscillation. Do you have any evidence that the internal side is still oscillating when the external side is not?

2. There are numerous points where the English could be improved. Please proofread carefully. For example:

Line 15 “domain, could not be achieved” -> “domain, it could not be achieved”

Line 18 “strength” -> “the strength”

Line 19 “However, strong vortices near the exit region of the fluidic oscillator lead to an almost non-symmetric pattern were created.” -> “However, strong vortices were created near the exit region of the fluidic oscillator, leading to an almost non-symmetric pattern.”

Line 20 “proper” -> “the proper” (or you can remove the word ‘method’).

Line 22 “most” -> “the most”

Line 30 “Multitude” -> “A multitude”

Line 36/37 Awkward sentence. Consider rewriting. The singular form of the verb should be used since the subject of the sentence is singular.

Line 38 “wide range” -> “a wide range”

Line 40 “range” -> “ranges”

Line 49 “double” -> “a double”

Line 68 “is alleged” -> “it is alleged”

Line 72 “increase” -> “increases”

Line 89 “be melt” -> “melt”

Line 90 “used” -> “it is used”

Line 116 “by developed software by our group” -> “by software developed by our group”

Line 175 “by increment” -> “by incrementing”

Line 176 “jet is inclined to attach one side” -> “the jet is inclined to attach to one side”

Line 182 “This causes that the jet is more pulled toward” -> “This causes the jet to be pulled toward”

Line 184 “further shrink” -> “further shrinking”

Lines 182-189 – Some sentences are awkward. Consider rewriting.

Line 237 “figs” -> “figures”

Line 250 “free” -> “the free”

Line 252 “fully” -> “the fully”

Line 254 “On the contrary, this ratio for free external domain” -> “In contrast, this ratio for the free external domain”

Line 256 “contour” -> “contours”

Line 256 “large” -> “the large”

Line 257 “free” -> “the free”

Line 258 “fully” -> “the fully”

Line 297 The sentence beginning “Due to …” is hard to understand. Consider rewording it.

Line 300 “compare” -> “compared”

Line 326 “Third” -> “The third”

Line 409 “cause” -> “causes”

Line 410 “free” -> “the free”

Line 413-415 – Awkward sentence. Consider rewording.

Line 416 “is exist” -> “exists”

Line 419 “fully” -> “the fully”

Reviewer 2 Report

The paper looks interesting and can add to the existing literature in this field. Here are my comments:

Did the authors consider plotting the relationships between other established nondimensional parameters such as the Strouhol number versus the Re number?

Did the authors consider the effects of geometric parameters such as the width of the channel on the oscillations?

There are typos on the figures. Please check throughout.

Polish English. There are occasional grammatical errors and typos.

Round 2

Reviewer 2 Report

It looks like the review's comments/concerns were answered, except for the typos in the figures. See the attached screenshots. There might be issues when the word is being converted into the pdf. But, it looks unprofessional.

Also, please address the following as well. I will not require further review. 

  • Mention the constant St number in the conclusions as well.
  • Please go through the whole manuscript to correct minor English errors and/or typos one more time.
